# Carbon Nanofiber Double Active Layer and Co-Incorporation as New Anode Modification Strategies for Power-Enhanced Microbial Fuel Cells

**DOI:** 10.3390/polym14081542

**Published:** 2022-04-11

**Authors:** Nasser A. M. Barakat, Mohamed Taha Amen, Rasha H. Ali, Mamdouh M. Nassar, Olfat A. Fadali, Marwa A. Ali, Hak Yong Kim

**Affiliations:** 1Chemical Engineering Department, Faculty of Engineering, Minia University, El-Minia 61519, Egypt; rashahussien22@yahoo.com (R.H.A.); mamdouh_nassar@yahoo.com (M.M.N.); olfat.fadali@yahoo.com (O.A.F.); marwa_moustafa2003@yahoo.com (M.A.A.); 2Microbiology Department, Faculty of Agriculture, Zagazig University, Zagazig 44511, Egypt; mtahaamen@gmail.com; 3Department of Nano Convergence Engineering, Jeonbuk National University, Jeonju 54896, Korea; khy@jbnu.ac.kr; 4Department of Organic Materials and Fiber Engineering, Jeonbuk National University, Jeonju 54896, Korea

**Keywords:** electrospinning, microbial fuel cells, carbon nanofibers, double layer, Co-doped

## Abstract

Co-doped carbon nanofiber mats can be prepared by the addition of cobalt acetate to the polyacrylonitrile/DMF electrospun solution. Wastewater obtained from food industries was utilized as the anolyte as well as microorganisms as the source in single-chamber batch mode microbial fuel cells. The results indicated that the single Co-free carbon nanofiber mat was not a good anode in the used microbial fuel cells. However, the generated power can be distinctly enhanced by using double active layers of pristine carbon nanofiber mats or a single layer Co-doped carbon nanofiber mat as anodes. Typically, after 24 h batching time, the estimated generated power densities were 10, 92, and 121 mW/m^2^ for single, double active layers, and Co-doped carbon nanofiber anodes, respectively. For comparison, the performance of the cell was investigated using carbon cloth and carbon paper as anodes, the observed power densities were smaller than the introduced modified anodes at 58 and 62 mW/m^2^, respectively. Moreover, the COD removal and Columbic efficiency were calculated for the proposed anodes as well as the used commercial ones. The results further confirm the priority of using double active layer or metal-doped carbon nanofiber anodes over the commercial ones. Numerically, the calculated COD removals were 29.16 and 38.95% for carbon paper and carbon cloth while 40.53 and 45.79% COD removals were obtained with double active layer and Co-doped carbon nanofiber anodes, respectively. With a similar trend, the calculated Columbic efficiencies were 26, 42, 52, and 71% for the same sequence.

## 1. Introduction

Researchers are motivated to investigate alternate sources of water and energy due to a shortage of both. In this regard, renewable energy generation from industrial and municipal wastewaters with a simultaneous treatment is an attractable new approach. Microbial fuel cells (MFCs) are a fantastic device that can be exploited to achieve this task, producing electrical energy from wastewaters with a simultaneous treatment. MFCs are an innovative environmental and energy system that converts organic pollutants in wastewater into electrical energy [1]. Modification in fuel cells could enhance the performance of the MFC through different ways such as anode or cathode modification [2,3], optimization of operating conditions [4], membrane improvement [5], and electrolyte modification [6]. Researchers have also worked to discover the optimum conditions to operate MFCs by applying various types of microorganisms, [7,8] media (fuel), [7,9] electrode materials/sub materials, [10,11] cell configurations, [12,13], and membranes [5]. However, among the aforementioned factors affecting the MFC performance, anodes have attracted the most attention so numerous reports have been published on anode modification compared to the other factors. 

Electrospinning is a flexible and practical method for producing ultrathin fibers. The advancement of electrospinning processes and the creation of electrospun nanofibers to suit or enable several applications have made remarkable progress. Therefore, many researchers have modified the performance of MFCs by utilizing the electrospinning technology [14,15]. As an anode material, carbon owns almost all the basic requirements such as high electrical conductivity, excellent chemical stability, biocompatibility, and low cost. Consequently, electrospun carbon nanofibers are the most utilized material to enhance MFC performance using the electrospinning technology.

Recently, pristine electrospun carbon nanofiber electrodes are colonized by exoelectrogenic strains—either the model bacterium *Shewanella oneidensis* or a wastewater-occurring bacterial consortium—and are then integrated into a lab-scaled setup [16]. Honestly, the nano pores existing in the carbon nanofiber mats are inconvenient for micro-scale biocatalyst species, microorganisms. In other words, penetration of the microorganisms through the carbon nanofiber anode is not an easy task, which trifles the nanostructure advantage and cancels the large surface area characteristic. Therefore, other strategies have been invoked to provide additional features to the electrospun carbon nanofiber-based anodes. For example, carboxylated multiwalled carbon nanotubes/carbon nanofibers composite electrode was fabricated by electrospinning and used as a hybrid anode to improve the MFC performance based on improving the cell attachment and decreasing the anode potential [17]. For the same purpose, the TiO_2_ (rutile)/carbon nanofiber composite anode was investigated in a MFC [18]. However, the generated powers from these trials were not satisfactory. Layered carbon fiber mats, prepared by layer-by-layer electrospinning of polyacrylonitrile onto thin natural cellulose paper and subsequent carbonization, have been introduced as a different strategy to enhance MFC performance [19]. Although, the layers thickness were designed to be very thin (in microns), cell penetration did not carry out through the inner layers due to the small pore size. 

The use of metal deposition to modify electrode materials is a recent clever method for improving the anode surface and, as a result, the MFC performance [20,21]. The selected metal must have high biocompatibility with bacteria, aid in increasing the electron transfer rate, and boost electrical conductivity, all of which promote the microorganism’s adhesion to the anode surface. Various metals have been used to modify the surface of carbonaceous materials, particularly the “essential” heavy metals (e.g., Co, Fe, Zn, Cu, and Ni), which have the capacity to improve the anode surface’s electrochemical activity as well as its bioactivity [22,23]. Cobalt is the most promising metal among the studied metals because of its potential to encourage micro growth and expedite micro cell adherence on the anode surface as well as its demonstrated high efficiency for power production. However, the metal content should be very small because a high amount might have a toxicity effect. Moreover, indirect contact with the microorganisms is recommended [24,25]. Aside from the bio-catalytic activity, cobalt can enhance many catalytic reactions such as the production of CNTs from acetylene [26]. 

The low performance of the single carbon nanofiber anode was further experimentally proven in this study. Therefore, in this report, we planned to introduce two novel approaches to enhance the performance of the carbon nanofiber mat as an anode in the MFC. First, two active layers of pristine carbon nanofibers were used. At the beginning, activation of the first layer was done by utilizing a single carbon nanofiber mat as the anode in the MFC for 24 h to create an active biofilm. Later on, another layer was installed over the first one to make a double active layer anode. The second approach was based on utilizing Co-incorporated carbon nanofibers that could be prepared according to our previous studies [27,28] as an anode in the MFCs. The used anolyte was industrial wastewater obtained from a food processing factory in Jeonju, South Korea. The results are interesting as the two approaches revealed better performances compared to the conventional carbon paper and carbon cloth electrodes. 

## 2. Materials and Methods

### 2.1. Materials

Carbon cloth (CC) and carbon paper (CP) were purchased from Electro Chem. Inc., Woburn, MA, USA. Cation exchange membrane (CEM, CMI-7000) was obtained from Membrane International Inc., Ringwood, NJ, USA. Polyacrylonitrile (PAN, Mwt 500,000, Sigma Aldrich, St. Louis, MO, USA), cobalt acetate tertrahydrate (CoAc, Sigma Aldrich), and N,N-dimethylformammide (DMF, SamChun Chem. Co., Ltd., Pyeongtaek, Korea) were utilized to prepare the pristine and Co-incorporated carbon nanofiber anodes. Wastewater from a food company located in Jeonju City, South Korea was utilized as the anolyte after a simple filtration process using filter paper to remove the solid particles. 

### 2.2. Anode Preparation

To prepare a 10 wt.% solution for the electrospinning process, a certain amount of polyacrylonitrile (PAN) was dissolved in N,N-dimethylformamide (DMF) by stirring at 60 °C for 8 h. A 0.1 mm diameter needle with a needle-collector distance of 17 cm was used to electrospin the homogeneous solution at ambient temperature under a 15 KV electrical field. Co-incorporated carbon nanofibers were prepared by dissolving a pre-calculated cobalt acetate precursor in the DMF to prepare a final CoAc/PAN/DMF solution containing 1% CoAc with respect to PAN. Stabilization of the initial electrospun nanofibers was performed by heating under an air atmosphere at 250 °C for 1 h, and then the graphitization process was performed at 900 °C under a nitrogen atmosphere for 1 h. Heating rate in both cases was fixed at 2.5 deg/min. 

### 2.3. MFC Construction and Operation

The use of an air cathode qualified the MFC to be a cost effective and portable device. An air cathode was prepared from carbon felt (2.5 cm × 2.5 cm, 3.18 mm, Alfa Aesar) and Pt/C (20%, Alfa Aesar, Haverhill, MA, USA), according to previous reports [29]. The hydrophobic carbon layer was faced to the air side to control the oxygen diffusion, while the other side of the carbon felt, which was loaded by Pt/C particles (0.5 mg/cm^2^), faced the water side. A cation exchange membrane (CEM) was employed as a proton exchange membrane. First, the CEM was treated by immersing it in a 1 M NaCl solution for 12 h at room temperature, then storing it in distilled water until it was needed. As illustrated in Figure 1, the cathode was attached to the membrane and positioned on one side of the wastewater-containing chamber followed by the anode. The membrane electrode assembly was sandwiched between two high corrosion resistance stainless steel current collectors. The solution volume in the anode chamber was 84 mL. For the double active layer anode, first the MFC was assembled using a single layer carbon nanofiber mat and left until the anode potential became almost stable, which was performed after around 24 h. Later, the cell was disassembled to insert a second carbon nanofiber layer over the present one before it was reassembled. Before feeding, the anolyte was purged by nitrogen gas for 5 min. 

### 2.4. Characterization

Using a scanning electron microscope (SEM JSM-IT200, JEOL, Japan), the surface morphology of the anode material was examined before and after the MFC work was completed. The electrodes were dried at 50 °C for roughly 1 h after the tests and then utilized for SEM examination. The crystal structure was examined by XRD characterization using a Rigaku X-ray diffractometer (XRD, Rigaku, Tokyo, Japan) with Cu Kα (λ = 1.540 Å). The internal structure was examined by transmission electron microscope (TEM, JEOL, Tokyo, Japan) at 200 kV. 

Linear sweep voltammetry (LSV) was carried out using VersaStat4 Potentiostat (AMETEK Scientific Instruments, Oak Ridge, TN, USA) to obtain the polarization curves at a scan rate of 1 mV/s using a two-electrode setup with the cathode as a working electrode and the anode as both the counter and reference electrodes. The greatest potential charge may be acquired if all substrates can be digested by the microorganisms to create current, and the Columbic efficiency (*CE*) is defined as the proportion of the total charge that is actually transferred to the anode from the substrate. The total charge obtained was calculated by integrating the current over time and the elimination of chemical oxygen demand (*COD*) in the MFC using the equation below [30]:(1)$$CE=\frac{M\int_{0}^{t}Idt}{FbV_{an}\Delta COD}$$
where *M* is the molecular weight of oxygen (32); *F* is Faraday’s constant; *b* = 4 indicates the number of electrons exchanged per mole of oxygen; *V_an_* is the volume of the liquid in the anode compartment; and *COD* is the change in the chemical oxygen demand (*COD*) over time *t*. In a steady state, *CE* = *MI/Fb V_an_ COD*. 

## 3. Results and Discussion

### 3.1. Anode Characterization

Precursor is the raw ingredient that is utilized to create carbon nanofibers (CNFs). PAN, polyphenol, viscose rayon, cellulose phosphate, phenolic, and pitch-based fibers are only a few of the synthetic and natural precursors used to make CNFs [31]. CNFs have also been synthesized using a variety of biomass resources. Flax fiber, oil palm fiber, cotton fiber, and lignin are examples of natural resources [32]. The major properties of the precursors utilized to manufacture the CNFs are their ease of conversion to CF, high carbon yield, and cost-effective processing. Acrylic precursors are chosen by CNFs makers in the industrial sector. As it is already proved in research and industrial levels, PAN is the most extensively employed acrylic precursor in the production of CNFs [33,34]. 

It was proved that heating of specifically cobalt and nickel acetates in an inert environment causes anomalous breakdown of the acetate anion to create reducing gases (carbon monoxide and hydrogen), resulting in the production of pure metal rather than the expected metal oxide form. Formation of pure cobalt was ascribed according to the following equations [27,35]:

Co(CH3COO)_2_·4H_2_O 🠖 Co(OH)(CH_3_COO) + 3H_2_O + CH_3_COOH
(2)


Co(OH)(CH_3_COO) 🠖 0.5CoO + 0.5CoCO_3_ + 0.5H_2_O + 0.5CH_3_COCH_3_
(3)


CoCO_3_ 🠖 CoO + CO_2_
(4)


CoO + CO 🠖 Co + CO_2_
(5)


Figure 2 displays the XRD pattern for the prepared Co-containing carbon nanofibers. Cobalt metal is present in the investigated powder, according to the pattern obtained. The development of cubic crystalline cobalt is indicated by the strong diffraction peaks at 2theta values of 44.35°, 51.65°, 75.95°, 92.35°, and 97.75°, which correspond to the (111), (200), (220), (311), and (222) crystal planes, respectively. The major grain size was determined to be around 19 nm using Scherrer’s equation. At room temperature, the two cobalt phases, face-center-cubic (FCC) and hexagonal close-packed (HCP), generally coexist and are difficult to distinguish from one another. The structure of the synthesized cobalt was identified as FCC cobalt using the JCPDS database (JCDPS, card no 15-0806). Furthermore, the wide peak at 26.3° corresponded to a 3.37 Å experimental d spacing, confirming the existence of graphite-like carbon (crystal plan (002), JCPDS; 41-1487). The inset displays the TEM image of the produced CNFs. As shown, the black spots point to highly crystalline parts in the nanofiber matrix, so these spots depict the cobalt nanoparticles incorporated inside the CNFs. Therefore, the final structure of the produced material can be explained as Co-incorporated CNFs. It is worth mentioning that both pristine and Co-incorporated CNFs display smooth and long CNFs, as clearly displayed in the SEM images (Figure 3). 

### 3.2. Single and Double Active Layer Anode Performance

The generated power not only from MFCs but also other kinds of fuel cells is the main criterion evaluating the performance of these energy producing devices. In contrast to the traditional fuel cells, the catalysts (electrogens) in the MFC layer is formed in-vivo on the surface of the anode during the preparation of the cell. Therefore, the anode total surface area is a highly controlling factor for the electrode performance. To enlarge the surface area, additional internal areas (e.g., pores) are created to maximize the number of the attached microorganisms. Indeed, CNFs possess the other required characteristics for the optimum anode such as excellent electrical conductivity, biocompatibility, and distinguished chemical stability. However, the nanopores in the CNF mats are insufficient to pass through or host the microorganisms. Therefore, the main advantage (the high surface area) of the nanostructure is not workable. Consequently, the apparent surface area will only be utilized to attach the microorganisms. The aforementioned hypothesis was proven experimentally in Figure 4. The expectation of the difficulty of the microorganisms’ penetration through the CNF mat was confirmed by the SEM analysis of the used single layer CNF anode (Figure 4). As shown in the figure, the microorganisms were too big to be hosted inside the CNF pores; instead, they attached to the outer surface. Therefore, the relatively good performance of the utilized single layer CNFs (Figure 5A) can mainly be attributed to the excellent electron transfer ability. The generated electrons in the microbial fuel cells were obtained from the metabolism of the organic pollutants in the microorganisms. Consequently, the generated current density directly proportions with the number of the attached microorganisms on the anode surface. Compared to CNFs, the carbon papers and carbon cloth possess higher porosity. However, as shown in Figure 5B,C, which displays the SEM image of the used carbon paper and carbon cloth anode, respectively, the electrodes’ bio-characteristics were not good enough to attract numerous microorganisms. In other words, although carbon paper and carbon cloth had larger porosity compared to the proposed CNF mat, the latter attracted more microorganisms to be attached on the surface due to its good biological properties. Nevertheless, in the case of CNFs, the microorganisms cannot penetrate to the inner layers. Consequently, the double CNF layer-based anode can have a better performance due to embedding numerous microorganisms.

Figure 5 displays the polarization and generated power density versus the current density for single CNF layer-assembled MFCs at different times. The generated power density was estimated by multiplication of the current with the corresponding cell potential divided by the anode surface area. In fact, the generated power densities and observed cell potentials were satisfactory compared to other reports due to the distinct electron transfer process of the nanofibrous morphology [36,37]; however, more improvement could be made if the active surface area could be enhanced or the surface properties were improved.

The performance of the double active layer CNF anode is represented in Figure 6, which displays the relationship between the generated power density and the cell potential with the current density. To properly introduce the advantage of the double active layer, the main cell parameters of the single and double active layer CNF-based MFCs are summarized in Figure 7. 

The experiments were repeated three times; the bars in Figure 7 are data points that represent the obtained errors. As shown in Figure 7A, which illustrates the influence of batching time on the generated powers in the two cells, at 24 h, the generated power density jumped from 9.7 ± 1.1 to 92.3 ± 2.5 mW/m^2^ due to the use of the second CNF layer. Moreover, for the double active layer, the power density reached 133 ± 3 mW/m^2^ after 120 h working time. The gradual decrease in the power density after the maximum value can be inputted to the mass transfer limitations. In other words, passing the substrates to the inner active layer faces mass transfer resistance. On the other hand, for the single layer CNF-based MFC, the generated power almost increased linearly with the batching time due to the availability of the substrates around the microorganisms. It is proposed that this problem facing the double active layer CNF-based cell can be fixed if agitation of the anolyte is performed.

Current reflects the collected electrons from the electrogen microorganisms attaching to the anode. As shown in Figure 7B, the current densities’ behavior resembles the power density attitude. Using the double active layer resulted in increasing the attached microorganisms, which caused an increase in the produced current. Numerically, after 24 h batching time, the detected current density was duplicated 10 times due to using two active layers as it increased from 25.7 ± 1.2 to 247.7 ± 4.2 mA/m^2^. However, at the maximum value (at 120 h), an almost 3-fold increase was observed as the detected current densities were 133.9 ± 3 and 474.4 ± 6.5 mA/m^2^ for the single and double active layer CNF-based MFC, respectively. 

Open circuit voltage (OCV) represents the maximum cell voltage that occurs at zero current density (open circuit). At the OCV state, the electrons are accumulated on the membrane of the microorganisms waiting the release moment. Therefore, the maximum number of electrons at the cell membranes and consequently the minimum anode potential are implemented at the OCV. When the cell is closed, the electrons start to pass through the outer circuit (the load) so the cell potential decreases.

Figure 7C depicts the impact of utilizing a double active layer anode on OCV at different batching times. As shown in the figure, for the double active layer CNF-based MFC, the OCV is directly proportionate with the time; it started as 670 ± 6.5 mV (at 24 h) and reached 821 ± 7.8 mV (at 172 h). For the single layer anode, at 24 h, the observed OCV was 532 ± 4.5 mV then decreased slightly and stabilized at 490 ± 4 mV at a batching time range of 72~96 h. Later, the OCV increased and became stable at 643 ± 6.5 mV after a 144 h working time until the end of the experiment (168 h). During the open circuit period, the adhesion forces between the microorganisms and the anode surface are weak, so microorganism release can take place. Therefore, the number of the microorganisms on the anode surface is in a dynamic equilibrium between the microorganisms’ attach and release rates. Accordingly, the small decrease in the OCV in the case of the single layer CNF-based MFC can be explained as increasing the washing out rate, so some microorganisms are released from the anode surface, and the released number of microorganisms is considerable with respect to the total number of the attached microorganisms, which was translated as a decrease in the OCV. The attaching rate increase may be due to the attachment of different kinds of microorganisms. On the other hand, in the case of the double active layer CNF anode, plenty of microorganisms were prisoned on the surface of the inner layer, so the number of released microorganisms was negligible compared to the total number; consequently, an almost linear increase in the OCV was observed. 

### 3.3. Co-Incorporated CNF Anode Performance

Heavy metals such as cadmium, lead, and mercury have received the most attention in terms of environmental protection, but other micronutrients also require consideration as well due to the likelihood of large soil loading. Among the elements present in amounts exceeding trace values in the soil, water, and air environments, cobalt (not yet fully known as a trace element) has attracted special attention [37,38]. Trace elements, which include cobalt, are among the numerous chemicals that have an impact on the course of microbiological activities. Maintaining appropriate levels in microorganism cells can help metabolic activities or boost vitamin B12 levels, which can help with growth [39]. The presence of cobalt in xylose isomerase enables the appropriate course of sugar metabolism in microorganisms, as demonstrated by the findings of Hlima et al. [40]. Because vitamin B12 production is required for the course of redox processes and nucleoprotein synthesis, cobalt is found in the coenzymes of *Rhizobium* bacteria or free-living *Azotobacter* bacteria, which are responsible for binding nitrogen from the air [41]. However, aside from the content limitation, a high amount of this metal can have a negative impact on the microorganisms, so the direct contact of the zero valent cobalt with the aqueous solution in an electrochemical device is highly not recommended due to the dissolution possibility of the metal, especially if it is a part of the anode. In other words, because cobalt is an active metal and is located at the anode, its ionization (i.e., cobalt can be oxidized and liberates electrons) in the MFC is highly expected. Therefore, to obtain the advantage of the cobalt and protect it from ionization, sheathing it in a strong shell is required. Accordingly, incorporation of cobalt nanoparticles inside carbon nanofibers was our target (see the inset in Figure 2). 

Figure 8 demonstrates the power generation and the polarization curve of the examined MFC using Co-incorporated CNFs as the anode after a 24 h batching time. As shown in the figure, a 121 mW/m^2^ power density was obtained, which was 32% more than that obtained from the double active layer CNF anode. Moreover, the observed current density from this MFC (397 mA/m^2^) was 60% more than that obtained from the double active layer CNF-based MFC at 249 mA/m^2^. Interestingly, the OCV of the Co-incorporated CNF–based cell was less than that observed from the double layer CNF one at 631 and 680 mV, respectively. The last finding about the OCV further draws attention to the advantage of cobalt incorporation. Lower OCV with higher power and current densities compared to the double active CNF-based MFC indicates that although the number of attached cells was lower, the good properties of the anode resulted in improving the power and current densities. 

### 3.4. Comparison with Carbon Cloth and Carbon Paper

Carbon cloth and carbon papers are standard materials widely used to evaluate the performance of proposed electrode materials. In this study, these materials were used as anodes in the same configuration of the utilized MFC. Comparison with the introduced anodes was first established in terms of the cell basic parameters, power and current densities, and OCV. Figure 9 depicts the obtained cell parameters obtained after a 24 h batching time. As shown in the figures, the proposed modification strategies showed better performances compared to the commercial anodes. As shown, the generated power densities were 121, 92, 52, and 68 mW/m^2^ for the Co-incorporated CNFs, double active layer CNFs, carbon paper, and carbon cloth anodes, respectively. This means that compared to carbon paper, the increase in power density was 132 and 77% when Co-incorporated CNFs and double active layer CNF anodes were used, respectively. Compared to carbon cloth, the increase in power density was 78 and 35% with the same sequence. In the same trend, as shown in Figure 9, the increase in the current density was 149 and 55%, and 77 and 10% for Co-incorporated and double active layer CNFs compared to the carbon paper and carbon paper, respectively. Although there was a distinct increase in terms of power and current densities, the results indicated that the surfaces of carbon paper and carbon cloths had more microorganisms compared to the proposed modified electrodes, which can be imputed to the higher active surface area of these electrodes than that of the modified ones. Accordingly, this increase in the number of the microorganisms was translated into enhancement in the OCV values. Numerically, the observed OCVs were 530, 680, 631, 789, and 802 mV for the single CNFs, double active layer CNFs, Co-incorporated CNFs, carbon paper, and carbon cloth-based MFCs, respectively.

The MFC main function is based on exploiting the microorganisms to degrade the organic pollutants in the wastewater, which results in generating electrons. Accordingly, the dual advantages of the MFC are generating electric energy with simultaneous treatment of the wastewaters. Therefore, from the standpoint of a wastewater treatment engineer, it is feasible to assess an MFC’s substrate conversion rate in terms of chemical oxygen demand (COD) by determining its COD removal efficiency or, better yet, its removal rate (thus taking into account the retention time of the substrate in the cell). 

Coulombic efficiency refers to the ratio of actual transferred electric charge to maximum value attainable if all of the substrate is removed to create a current, and is a crucial measure for evaluating MFC performance [42]. The MFC’s Coulombic efficiency, the ratio between electron moles extracted as current and total electron moles made accessible via substrate oxidation, is used to determine the global efficiency of the bioelectrochemical process [43]. 

Figure 10 displays the COD of the output wastewaters after utilizing the MFCs using different anodes. Initially, the COD of the used water was 950. As shown, the treatment efficiency depended on the used anode. Typically, the COD of the treated solutions (after 24 h) was 870, 565, 515, 673, and 580 when single CNFs, double active CNFs, Co-incorporated CNFs, carbon paper, and carbon cloth were used as the anodes, respectively. These data indicate a COD removal efficiency of 8.4, 40.5, 45.8, 29.2, and 38.9% for the aforementioned anodes with the same sequence. As seen, the maximum COD removal efficiency (45.8%) corresponded to the Co-incorporated CNFs, which sheds light on the advantage of cobalt incorporation. 

Along the same lines of COD removal behavior, Coulombic efficiency estimation results run in the same attitude. As shown in the figure, the maximum Coulombic efficiency (71%) was obtained with Co-incorporated CNFs while the lowest value belonged to a pristine single layer CNF layer anode of 6%. Moreover, as shown in the results, the double active layer CNF anode revealed better Coulombic efficiency compared to carbon paper and carbon cloth at 52, 26, and 42%, respectively. 

To properly evaluate the proposed anodes, a comparison with some reported MFC anodes in the literature is introduced in Table 1. As shown in the table, although the estimated power of the Co-incorporated CNF-based cell was after a 24 h batching time only, the used anode had an excellent performance compared to almost all the cited materials in Table 1. Similarly, the power generated (135 mW/m^2^) from the double active layer CNF-based MFC after 120 h was higher than the numerous reported values obtained from different materials. Finally, as aforementioned, although a single layer CNF anode did not possess a high active area, its performance was very satisfactory; the generated power density (after 168 h batching time) exceeded many introduced anodes in the literature. 

The obtained good results for the Co-incorporated CNF anode can mainly be attributed to its capacity to boost micro growth and accelerate micro cell adhesion on the anode surface as well as shown high efficiency for power generation [44]. From the electrical conductivity point of view, carbon nanofibers have very good electrical conductivity (4.2 S/cm) [45]. Compared to pristine CNFs, cobalt possesses very high electrical conductivity. However, since it is incorporated in the form of discrete nanoparticles along with the carbon nanofiber matrix, the produced composite has a relatively higher conductivity [46]. Therefore, it can be concluded that the performance improvement due to cobalt incorporation is mainly and partially imputed to the enhancement in the biological and physical properties of the used anode, respectively.

## 4. Conclusions

Polyacryonitrile polymer is a very good precursor to prepare good morphology carbon nanofibers; moreover, the addition of cobalt acetate to the initial electrospun solution does not affect the final morphology and results in producing Co-incorporated carbon nanofibers. The nanoscale pores existing in the pristine carbon nanofiber mat do not share in the performance when the mat is invoked as an anode in the microbial fuel cells. However, the excellent electron transfer process through the carbon nanofibers relatively compensate the negligible role of the nanopores, so compared to many reported anodes, a considerable performance was obtained when a single layer from carbon nanofibers is utilized as an anode in the microbial fuel cell. However, the performance can be enhanced when two adjacent microorganism-attached carbon nanofiber layers were used as the anode. Moreover, incorporation of cobalt nanoparticles in the carbon nanofibers can distinctly enhance the performance carbon nanofibers as anodes in the microbial fuel cells. Overall, this study introduces two novel strategies to overcome the low active surface area of the electrospun carbon nanofiber mats to be used as valuable anodes in the microbial fuel cells. 

## Figures and Tables

**Figure 1 polymers-14-01542-f001:**
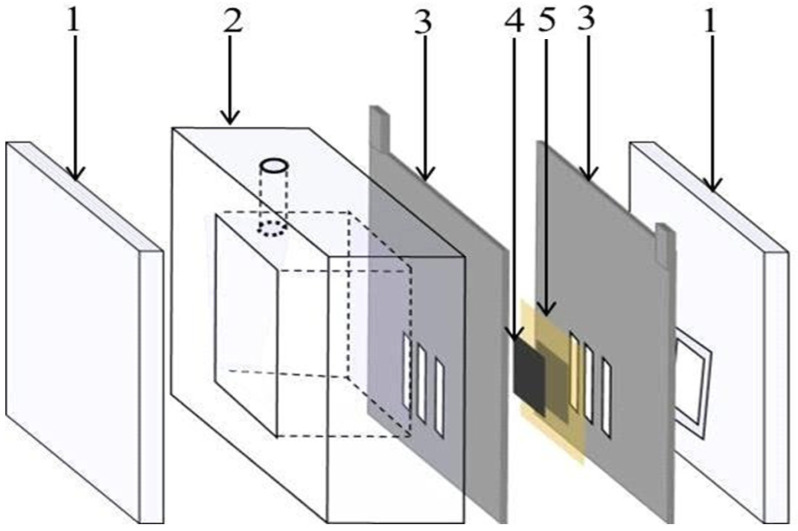
A schematic diagram of the cell structure: 1—Cover plates, 2—Anode chamber, 3—Current collectors, 4—Anode, and 5—Membrane and cathode.

**Figure 2 polymers-14-01542-f002:**
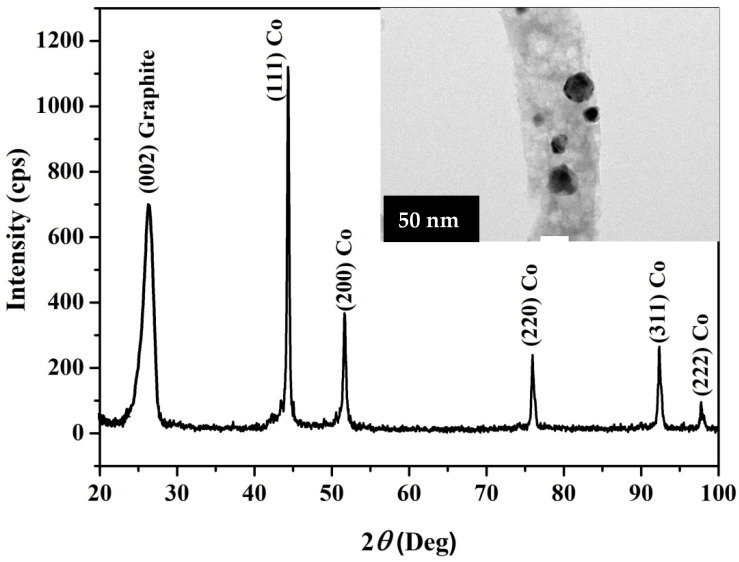
XRD pattern for the prepared Co-incorporated carbon nanofibers. The inset displays the TEM image of the prepared Co-incorporated carbon nanofibers.

**Figure 3 polymers-14-01542-f003:**
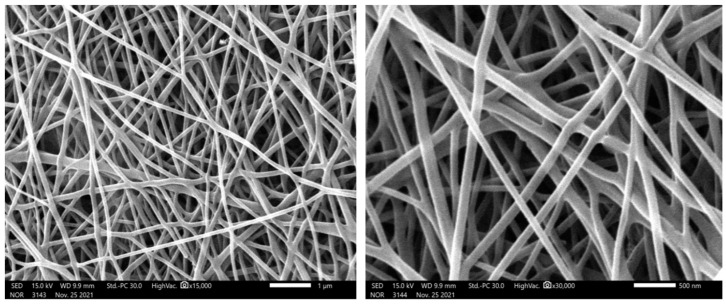
Two SEM image magnifications of the prepared carbon nanofibers.

**Figure 4 polymers-14-01542-f004:**
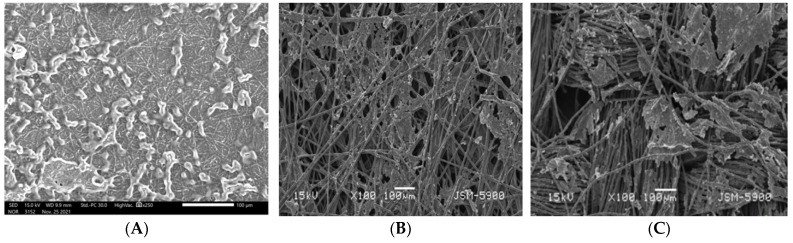
SEM image for the used carbon nanofiber: (**A**), carbon paper; (**B**) carbon cloth; (**C**) anodes in the microbial fuel cell.

**Figure 5 polymers-14-01542-f005:**
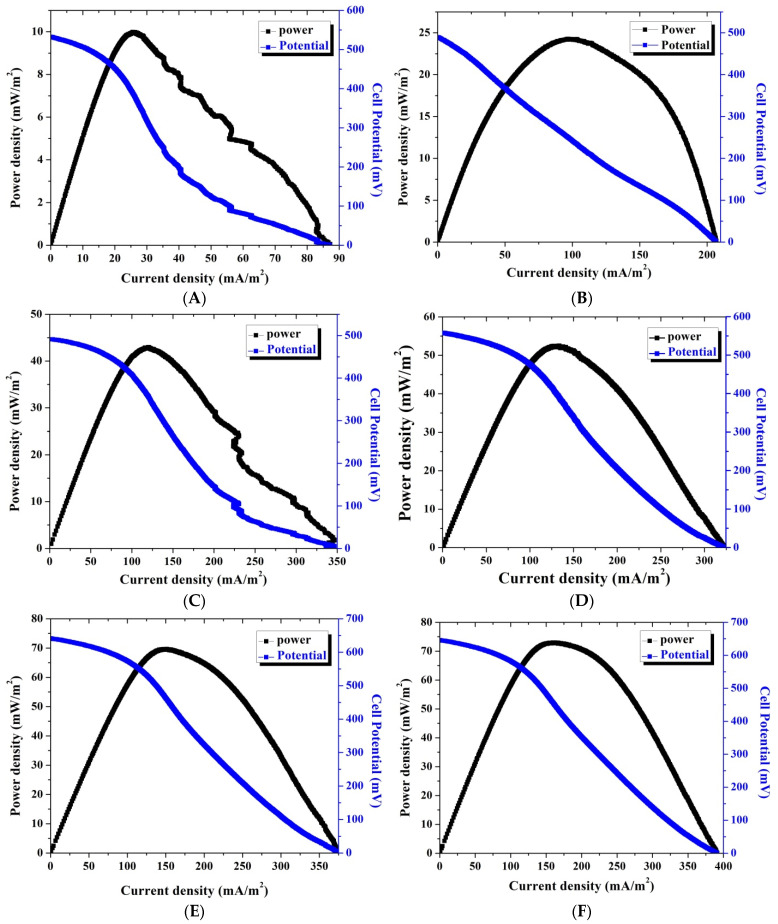
Polarization and power density curves of batch-mode and single-chamber MFCs using a single layer carbon nanofiber anode after 24: (**A**) 72; (**B**) 96; (**C**) 120; (**D**) 144; (**E**) and 168; (**F**) h batching time.

**Figure 6 polymers-14-01542-f006:**
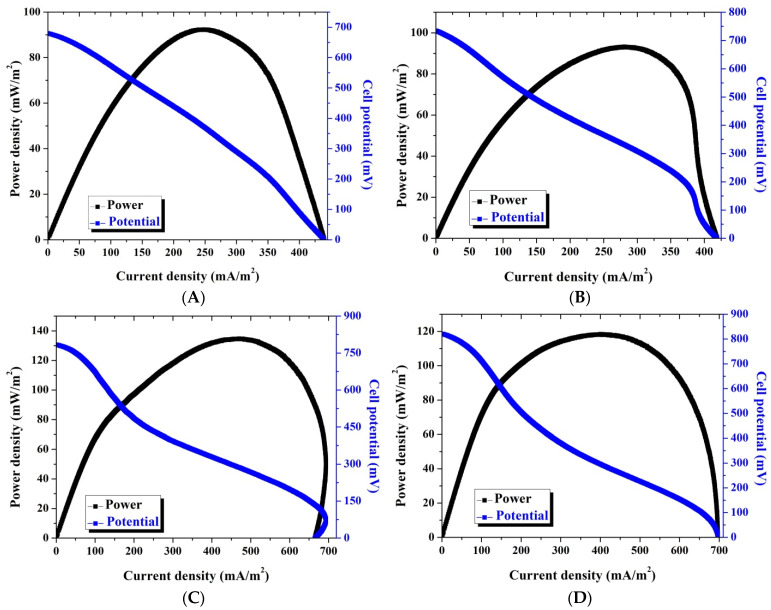
Polarization and power density curves of batch-mode and single-chamber MFCs using a double layer carbon nanofiber anode after 24: (**A**) 72; (**B**) 120; (**C**) and 172; (**D**) h batching time.

**Figure 7 polymers-14-01542-f007:**
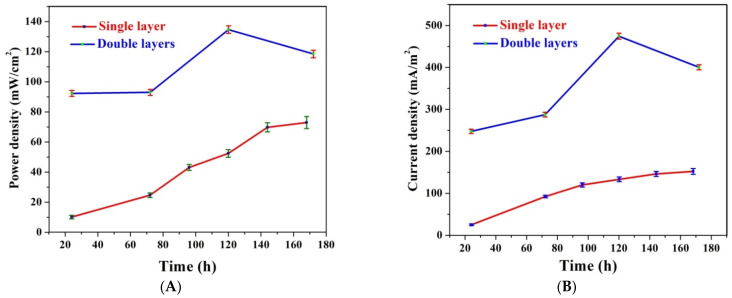
Influence of utilizing double layer carbon nanofibers as anodes on the obtained power density (**A**); current density (**B**); and open circuit potential (**C**).

**Figure 8 polymers-14-01542-f008:**
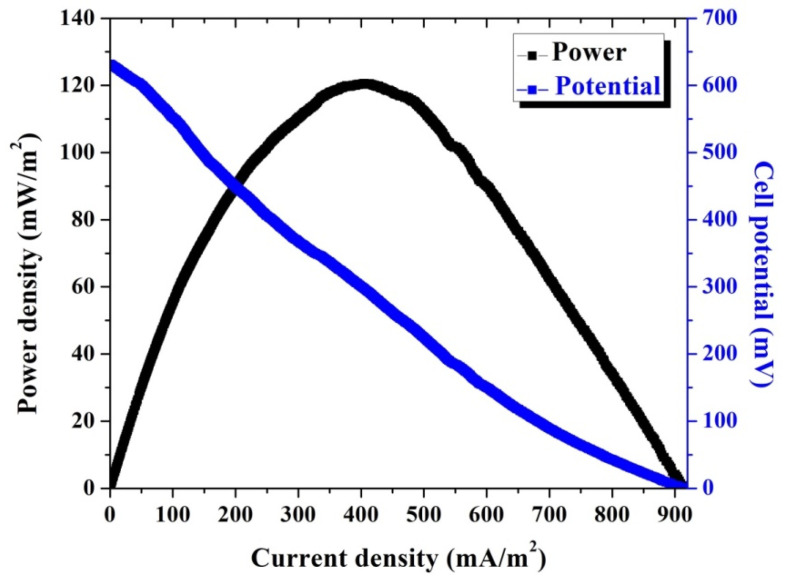
Polarization and power density curves of batch-mode and single-chamber MFCs using a cobalt-incorporated carbon nanofiber anode after 24 h batching time.

**Figure 9 polymers-14-01542-f009:**
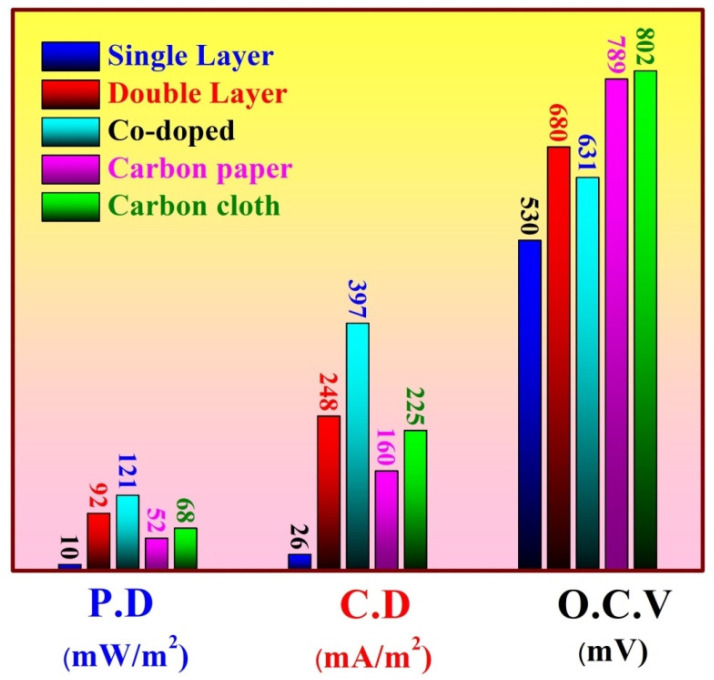
Power density, open circuit potential, and current density after 24 h batching of MFCs using different anodes.

**Figure 10 polymers-14-01542-f010:**
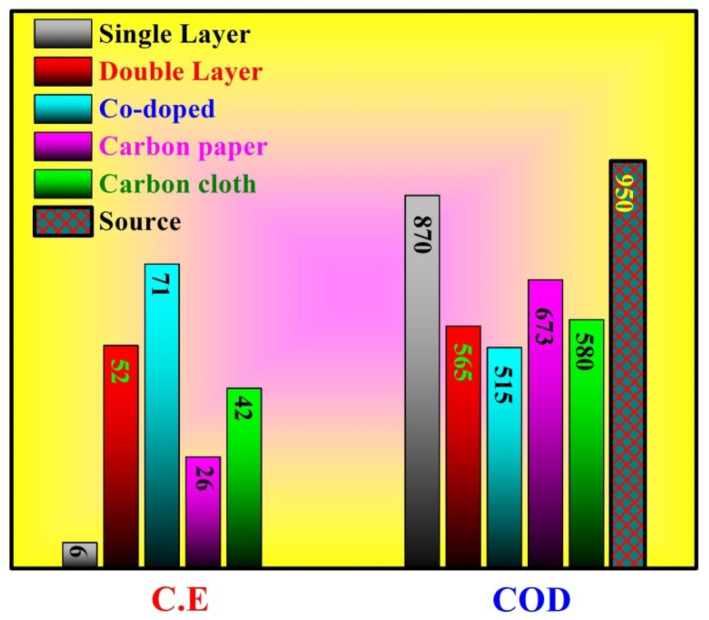
Columbic efficiency (CE) and COD of the final solution after 24 h batching of MFCs using different anodes.

**Table 1 polymers-14-01542-t001:** Performance of the reported MFC anodes in the literature in terms of power density generation compared with the proposed anodes in this study.

Cell Type	Microorganism Media	Anode Material	Power Density(mWm^−2^)	Improving (%)
Single CNFs	Co-CNFs	DoubleCNFs
Single chamber [47]	Local domestic wastewater	Graphite rods	26	181 	365 	419 
Mediator less MFC [48]	*P. aeruginosain*	Graphene-modifiedcarbon cloth	50	46 	142 	170 
A dual chamber fuel cell [49]	*Geobacteria*. sulfurreducens	Solid graphite	13.1	457 	824 	931 
Two-chamber flat plate mediator-less MFC [50]	*Shewanella putrefaciens* in Luria broth	Graphite plate	39.2	86 	209 	244 
Mediator-less [51]	*Saccharomyces cerevisiae* yeast	Carbon paper	3	2333 	3933 	 4400
Mediator [52]	*Proteus vulgaris* grow on Glucose	Glassy carbon	4.5	1522 	2589 	2900 
Two chamber [53]	Mixed consortium,Continuous grow on Sucrose	Granular graphite	47	55 	157 	187 
Mediator-less [54]	*Rhodoferax ferrireducens* grow on Glucose	Graphite foam	33	121 	267 	309 
Single chamber [55]	Mixed culture of microorganism utilize Acetate	Carbon paper	13	462 	831 	938 
Single chamber [55]	Mixed culture of microorganism utilize Butyrate	Carbon paper	7.6	861 	1492 	1676 
Mediator [56]	*Proteus vulgaris* grow on Glucose	Glassy carbon	9	711 	1244 	1400 
Mediator [57]	*Escherichia coli* grow on Lactate	Plain graphite	3.6	1928 	3261 	3650 
Mediator [57]	Activated sludge waste water mixed with Lactate	Woven graphite	34	115 	256 	297 
Two chamber [58]	*Pseudomonas aeruginosa* Glucose	Plain graphite	88	−17 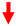	38 	53 
Single air type [51]	*Saccharomyces cerevisiae*	Carbon paper	3.2	2181 	3681 	4119 
Two chamber [59]	*Saccharomyces cerevisiae*	Graphite plate	4.9	1390 	2369 	2655 
Marine sediments [60]	Artificial marine	Stainlesssteel plate	23	217 	426 	487 
Two chamberCylindrical [61]	Anaerobic sludgebrewery wastewater	Reticulate vitreouscarbon packed	170	−57 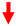	−29 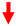	−21 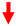
Dual chamberH-cell [62]	*Shewanella oneidensis*	Solid graphite	9.3	685 	1201 	1352 
Dual chamber [63]	*Pseudomonas aeruginosa*isolated from palm oil anaerobic sludge	Poly acrylonitrile carbon felt	107.35	−32 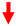	13 	26 
The dual-chambered [64]	Waste water	Carbon rods	78.25	−7 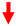	55 	73 
Open-air cathode [65]	*Saccharomyces cerevisiae* yeast	carbon paper modified with Co 30%	20	265 	505 	575 
Single air cathode [66]	Food waste water	Carbon paper	52	40 	133 	160 
Single air cathode [66]	Food waste water	Carbon cloth	68	7 	78 	99 
Single air cathode [66]	Food waste water	Graphite paper	175	−58 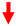	−31 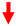	−23 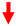
Single chamber air-cathode	Food waste water	Single CNF layer	73	0	66 	85 
Single chamber air-cathode	Food waste water	Double CNFs	135	−46 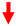	−10 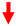	0
Single chamber air-cathode	Food waste water	Co-incorporated CNFs	121	−40 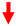	0	12


: enhancement; 
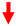
: decrement.

## Data Availability

The data presented in this study are available upon request from the corresponding author.

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
