# Peer review of "Carbon Nanofiber Double Active Layer and Co-Incorporation as New Anode Modification Strategies for Power-Enhanced Microbial Fuel Cells"

_polymers, 2022, doi:10.3390/polym14081542_

Round 1

Reviewer 1 Report

The manuscript (polymers-1615504) improves the performances of the carbon type anode of microbial fuel cell (MFC) by properly assembling carbon nanofibers and incorporating metal dopes. It is an interesting and potentially useful study. After reading the whole manuscript, I am wondering what is the basis for improving the performances of MFC. Is it lower anode resistance or better bio-compatibility? Could authors provide more data and discussions from the view of microorganisms? My other comments are listed below.

  1. Figure 2: The scale bar and texts in TEM image are too small to be seen clearly.
  2. Figure 5: I am wondering that whether the number of the microorganisms attached on anode surface affects current density? Is there data showing the distributions of microorganisms in anodes?
  3. Lines 316-325: Is there experimental data for the numbers of microorganisms on anode?
  4. Figure 9: Could authors show the morphologies of carbon cloth and carbon paper? How about the microorganisms attached on different carbon materials after 24h batching? How about the internal resistance of these anodes?

Author Response

Reviewer: 1

First, we strongly appreciate the valuable comments addressed by the respected reviewer which distinctly strengthened the manuscript. 

The manuscript (polymers-1615504) improves the performances of the carbon type anode of microbial fuel cell (MFC) by properly assembling carbon nanofibers and incorporating metal dopes. It is an interesting and potentially useful study. After reading the whole manuscript, I am wondering what is the basis for improving the performances of MFC. Is it lower anode resistance or better bio-compatibility? Could authors provide more data and discussions from the view of microorganisms?

Response: The obtained good results for the Co-incorporated CNFs anode can be mainly attributed to its capacity to boost micro growth and accelerate micro cell adhesion on the anode surface, as well as its shown high efficiency for power generation1. From the electrical conductivity point of view, carbon nanofibers have very good electrical conductivity (4.2 S/cm)2. Compared to pristine CNFs, cobalt possesses very high electrical conductivity. However, since it is incorporated in the form of discrete nanoparticles along with the carbon nanofiber matrix, the produced composite has a relatively higher conductivity3. Therefore, it can be concluded that the performance improvement due to cobalt incorporation is mainly and partially imputed to the enhancement in the biological and physical properties of the used anode, respectively.

This explanation was added in the revised manuscript.  

My other comments are listed below.

Figure 2: The scale bar and texts in TEM image are too small to be seen clearly.

Response: The figure has been updated

Figure 5: I am wondering that whether the number of the microorganisms attached on anode surface affects current density? Is there data showing the distributions of microorganisms in anodes?

Response: We would like to thank the respected reviewer about this good comment. SEM images for the used carbon cloth and carbon paper anodes were newly added in the revised manuscript with the following explanation.

The generated electrons in the microbial fuel cells are obtained from the metabolism of the organic pollutants in the microorganisms. Consequently, the generated current density directly proportions with the number of the attached microorganisms on the anode surface. Compared to CNFs, the carbon papers and carbon cloth possess higher porosity. However, as shown in figure 5B and figure 5C which display SEM image of the used carbon paper and carbon cloth anodes, respectively, the electrodes bio-characteristics are not good enough to attract numerous microorganisms. In other words, although carbon paper and carbon cloth have larger porosity compared to the proposed CNFs mat, the later attracts more microorganisms to be attached on the surface due to its good biological properties. Nevertheless, in the case of CNFs, the microorganisms cannot penetrate to the inner layers. Consequently, the double CNFs layer-based anode can have better performance due to embedding numerous microorganisms.        

Lines 316-325: Is there experimental data for the numbers of microorganisms on anode?

Response: We agree with the respected reviewer that estimating the number of the microbes can add a significant sense. However, due to COVID-19, we could not make this analysis due to lack of the required chemicals and/or facilities. We hope the respected reviewer accepted this excuse especially we used several measurements are directly related to the number of the attached microorganisms such as COD removal efficiency, Columbic efficiency, generated power density and the obtained current density.

Figure 9: Could authors show the morphologies of carbon cloth and carbon paper? How about the microorganisms attached on different carbon materials after 24h batching? How about the internal resistance of these anodes?

Response: Two SEM images were added to revised manuscript showing the surface morphology of the used carbon paper and carbon cloth anodes. Moreover, the anode resistances have been discussed as it was explained in the response about the general comment.  

Reviewer 2 Report

Carbon nanofiber anode was prepared by calcination of stabilized polyacrylonitrile electrospun nanofiber mat under nitrogen atmosphere. Co-doped carbon nanofiber mats could be prepared by addition of cobalt acetate to the polyacrylonitrile/DMF electrospun solution. The manuscript contains useful information and valuable results. The paper is publishable after minor modification.

  1. It seems that the Abstrct is a little bit long. It is suggested to focus to the imprtant results. A sligth shortening would be advicable.
  2. The metallic cobalt was reached from acetates. Figure 2 shows cobalt metal in XRD. It is well-known from the literature that it i shard to obtain metallic cobalt. Even the precursor is carbonyl, some Co ions remain in the structure of support. Please take the consideration of the recent work (Physical Chemistry Chemical Physics 15 (2013) 15917-15925). The metal loading has a significant role in Co/Co2+ ratio. CoO is a stable compound.
  3. Co/Al2O3 catalysts used for production of carbon nanotubes from acetylene by CCVD. Please consider this observation, perhaps it helps for interpretation (see: IEEE Transaction on Nanotechnology 3 (2004) 73-79).
  4. Is the carbonfiber structure and morphology stable after Co-incorporation?

Author Response

Reviewer: 2

Carbon nanofiber anode was prepared by calcination of stabilized polyacrylonitrile electrospun nanofiber mat under nitrogen atmosphere. Co-doped carbon nanofiber mats could be prepared by addition of cobalt acetate to the polyacrylonitrile/DMF electrospun solution. The manuscript contains useful information and valuable results. The paper is publishable after minor modification.

First, we would like to introduce our strong appreciation to the respected reviewer about his valuable time in evaluating our manuscript.  

It seems that the Abstract is a little bit long. It is suggested to focus to the imprtant results. A slight shortening would be advisable.

Response: The abstract has been updated.

The metallic cobalt was reached from acetates. Figure 2 shows cobalt metal in XRD. It is well-known from the literature that it i shard to obtain metallic cobalt. Even the precursor is carbonyl, some Co ions remain in the structure of support. Please take the consideration of the recent work (Physical Chemistry Chemical Physics 15 (2013) 15917-15925). The metal loading has a significant role in Co/Co2+ ratio. CoO is a stable compound.

Response: Thank you for this kind advice which was considered in the revised manuscript.

Co/Al2O3 catalysts used for production of carbon nanotubes from acetylene by CCVD. Please consider this observation, perhaps it helps for interpretation (see: IEEE Transaction on Nanotechnology 3 (2004) 73-79).

Response: Thank you for this kind advice which was considered in the revised manuscript.

Is the carbonfiber structure and morphology stable after Co-incorporation?

 Response: As can be seen in the TEM image, Co NPs are sheathed inside the carbon nanofiber matrix which enhances the anode stability because of isolating the metal from the aqueous solution.

Round 2

Reviewer 1 Report

The manuscript has been properly revised.